# Effects of a Brief E-Learning Resource on Sexual Attitudes and Beliefs of Healthcare Professionals Working in Prostate Cancer Care: A Pilot Study

**DOI:** 10.3390/ijerph181910045

**Published:** 2021-09-24

**Authors:** Eilís M. McCaughan, Carrie Flannagan, Kader Parahoo, Sharon L. Bingham, Nuala Brady, John Connaghan, Roma Maguire, Samantha Thompson, Suneil Jain, Michael Kirby, Seán R. O’Connor

**Affiliations:** 1Institute of Nursing and Health Research, Ulster University, Jordanstown BT37 0QB, UK; em.mccaughan@ulster.ac.uk (E.M.M.); c.flannagan@ulster.ac.uk (C.F.); ak.parahoo@ulster.ac.uk (K.P.); bingham-s4@ulster.ac.uk (S.L.B.); 2Northern Health and Social Care Trust, Antrim BT41 2RL, UK; nuala.brady@northerntrust.hscni.net; 3Department of Computer and Information Sciences, University of Strathclyde, Glasgow G11 XH, UK; john.connaghan@strath.ac.uk (J.C.); roma.maguire@strath.ac.uk (R.M.); 4Urology Department, Belfast City Hospital, Belfast BT9 7AB, UK; samanthae.thompson@belfasttrust.hscni.net; 5Centre for Cancer Research and Cell Biology, Queen’s University Belfast, Belfast BT9 7AE, UK; s.jain@qub.ac.uk; 6Clinical Oncology, Northern Ireland Cancer Centre, Belfast City Hospital, Belfast BT9 7AB, UK; 7Faculty of Health and Human Sciences, University of Hertfordshire, London AL10 9AB, UK; kirbym@globalnet.co.uk; 8The Prostate Centre, London W1G 7AF, UK; 9Centre for Public Health, Queen’s University Belfast, Belfast BT12 6BA, UK

**Keywords:** sexual wellbeing, prostate cancer, e-learning

## Abstract

Sexual issues and treatment side effects are not routinely discussed with men receiving treatment for prostate cancer, and support to address these concerns is not consistent across settings. This study evaluates a brief e-learning resource designed to improve sexual wellbeing support and examine its effects on healthcare professionals’ sexual attitudes and beliefs. Healthcare professionals (*n* = 44) completed an online questionnaire at baseline which included a modified 12-item sexual attitudes and beliefs survey (SABS). Follow-up questionnaires were completed immediately after the e-learning and at 4 weeks. Data were analysed using one-way, repeat measures ANOVAs to assess change in attitudes and beliefs over time. Significant improvements were observed at follow-up for a number of survey statements including ‘knowledge and understanding’, ‘confidence in discussing sexual wellbeing’ and the extent to which participants felt ‘equipped with the language to initiate conversations’. The resource was seen as concise, relevant to practice and as providing useful information on potential side effects of treatment. In brief, e-learning has potential to address barriers to sexual wellbeing communication and promote delivery of support for prostate cancer survivors. Practical methods and resources should be included with these interventions to support implementation of learning and long-term changes in clinical behaviour.

## 1. Introduction

Prostate cancer is the most common male cancer and long-term side effects associated with different treatment approaches are common [1,2]. In a recent large-scale survey, approximately 80% of men reported poor sexual function post-treatment [3]. Treatment guidelines [4,5] endorse delivery of psychosexual care for prostate cancer patients with recommendations made for the minimal level of support that should be provided. Despite evidence indicating that men with prostate cancer want healthcare professionals to discuss sexual issues and side effects of treatment [6,7,8], sexual aspects of recovery are often not discussed during post-treatment follow-up appointments and support is not provided consistently across settings [9]. Consequently, men with prostate cancer frequently report that they do not receive adequate information to manage sexual concerns [10,11]. This can be associated with increased psychological morbidity, including depression and relational dissatisfaction, as well as reductions in self-efficacy and overall quality of life [11].

Initiating discussions around sexual concerns in clinical practice can be challenging. A number of interpersonal factors have been identified as potential communication barriers. Healthcare professionals often regard patients’ sexual lives as being too personal to ask about [12]. Wider social influences and attitudes to sex and sexuality, including embarrassment, not being comfortable with the topic or not wishing to cause offence may also lead to active avoidance of the issue [12]. Healthcare professionals also report a perception that management of sexual issues is not within their professional role and that they feel unequipped to deal with sexual issues [13,14]. Other factors, including limited availability of onward referral services can also limit discussions further. Given their frequency and substantial impact [3], sexual concerns and potential side effects of treatment should be discussed routinely with all patients. Healthcare professionals have an essential role in ensuring patients’ sexual concerns are addressed and that appropriate support is provided, including onward referral to relevant services. There is therefore a clear need for approaches that provide healthcare professionals with the skills and capacity to routinely deliver sexual care and support. However, there is currently limited evidence exploring how communication can be enhanced and how conversations around sexual wellbeing can be supported in routine practice.

A study exploring evidence regarding knowledge and attitudes of oncology nurses and factors linked to provision of sexual support [15] concluded that continuing communication skills training education is needed to address assumptions around sexual issues that restrict sexual care communication in cancer care. Similar findings were found in a mixed-methods review exploring barriers to communication around sexual wellbeing in clinical practice that identified themes covering attitudes to sex and sexual wellbeing, patient factors, organisational factors, strategies to overcome barriers and training needs [16]. This review found healthcare professionals acknowledged the importance of discussing and providing support for sexual wellbeing needs, but recognised it is not routinely provided and highlighted a need for brief educational and support tools to promote effective conversations with patients.

For healthcare professionals to adequately address sexual wellbeing issues, they require an awareness of the impact of sexual issues on patients as well as knowledge and skills to effectively engage with patients and assess needs to provide appropriate evidence-based management [17]. In addition, they may require interventions designed to address attitudinal barriers to sexual wellbeing discussions in practice. Easily accessible, evidence-based e-learning modules can provide an approach by which all healthcare professionals in primary, and secondary care settings, who provide support for patients diagnosed with, or receiving active treatment for prostate cancer can undertake foundation level training to ensure they are prepared to deliver essential information and support to patients. Such interventions have the potential to improve healthcare professional communication and patient-important outcomes [18]. As attitudinal factors can act as barriers to healthcare professionals’ communication regarding sexual wellbeing concerns in patients with prostate cancer, the aim of the study was to evaluate a brief e-learning resource entitled ‘talking about sex after prostate cancer’ and explore its effects on healthcare professionals’ attitudes and beliefs around sexual wellbeing and prostate cancer, and their perspectives on its use.

## 2. Materials and Methods 

### 2.1. Study Design

A pilot study design was used with evaluations at baseline and at two follow-up time-points (immediately after and 4 weeks after completion of the e-learning). A minimum sample of 43 participants was calculated in G*power [19] based on a one-way repeat measures ANOVA using a power of 0.95 and a small estimated effect size of 0.01. Ethical approval for the study was provided by the Office for Research Ethics Committees Northern Ireland (reference number: 17/NI/014).

### 2.2. Study Population

Participants were healthcare professionals working in the area of prostate cancer care. No exclusions were applied to professional group or years of clinical experience. Recruitment was via e-mail invitation, posters located in clinical areas and through social media messages. A link to the e-learning resource was also included on the healthcare professional online learning platform of Prostate Cancer UK. Written, informed consent was obtained from all participants who took part in the study.

### 2.3. Intervention Content

A systematic, iterative and theory-based approach was used to develop the content and structure of the e-learning resource. The development phase included establishment of an expert group consisting of men with prostate cancer, and healthcare professionals working in uro-oncology and primary care settings; conduct of evidence reviews [16] and qualitative data synthesis from semi-structured interviews and focus group discussions with end users and field content experts. These were used to identify core or essential elements of the resource. In addition, content and healthcare professional perspectives on feasibility and acceptability were also evaluated at a 2-h facilitated workshop attended by 21 clinical nurse specialists that included small group discussions and demonstrations. After building initial versions of the resource, modifications were made based on usability testing and further rounds of qualitative interviews. Following this, a final version of the resource was built as a SCORM package integrated into a moodle open-source e-learning platform (https://talkingaboutsex-cancercare.org/course/view.php?id=2, accessed on 6 August 2021). A number of interactive elements and reflective activities are included in the resource such as a ‘virtual tutor’, quiz questions, videos showing patient and partner interviews, conversation demonstrations and goal-setting activities.

The resource consists of three sections: (1) an introduction and background to the area, (2) a framework for structuring sexual wellbeing discussions in practice and (3) methods with which to integrate learning into routine practice. Section one is aimed at raising awareness around the impact of prostate cancer on sexual wellbeing, including the effects of different treatment options. It also discusses key barriers preventing sexual wellbeing concerns being addressed in men with prostate cancer (from a societal and health system perspective, as well as patient and partner, and healthcare professional perspectives). Section two introduces an Engagement, Assessment, Support and Signposting (EASSi) framework as a means of addressing key barriers to routine engagement around sexual wellbeing with patients in the clinical setting. This framework is aimed at providing a mechanism to ensure healthcare professionals have the knowledge and skills to effectively engage with and support men and their partners. Of the four components included in the tool, ‘Engagement’ is focused on ensuring routine sexual wellbeing discussions take place with all men, acknowledging that such sexual issues are not easy to discuss and recognising that associated side effects of treatment can have a substantial impact. The ‘Assessment’ component includes questions on treatment type and relationship status to provide tailored support. The ‘Support’ component provides appropriate information on common sexual challenges (relevant to treatment and relationship status) and information on coping strategies. The ‘Sign-posting’ component provides other support options including online self-management and other resources specific to individual needs. Section three of the e-learning resource includes activities to support integration of learning into routine practice, including suggested goal setting activities and downloadable resources, including patient handouts and a simplified A4 poster version of the engagement framework to remind healthcare professionals to use the EASSi framework in routine practice.

### 2.4. Data Collection

The primary outcome measure was change in sexual attitudes and beliefs survey (SABS) scores. Good internal consistency (Cronbach’s alphas of 0.75 to 0.82) and test–retest reliability (r = 0.85; *p* < 0.001) of the survey has been demonstrated [20]. In the current study, we used a modified version of the SABS to ensure it related specifically to prostate cancer care. Modifications were based on pilot testing of the survey involving a sample of healthcare professionals who were not involved in the main study (*n* = 66). Changes made to the survey and internal consistency measures are described in Appendix A. The modified survey consisted of twelve statements including perceptions around sexual wellbeing knowledge and understanding and confidence in discussing sexual concerns with patients. Each question was rated on a four-point Likert scale based on the extent to which the participant agreed or disagreed with each statement (1 = Strongly Disagree; 4 = Strongly Agree). Five items are reversed during scoring. A total score ranging from 12 to 48 is assigned with lower scores indicating greater barriers to sexual wellbeing communication. Acceptable internal consistency of the modified survey was observed (Cronbach’s alpha: 0.69). Pre-test measurements were recorded prior to completion of the e-learning at Timepoint 0 (T0), immediately after completion at Timepoint 1 (T1) and at 4 weeks (Timepoint 2; T2). Additional questions were also included at Timepoint 1 and 2 exploring perspectives on usefulness and usability. E-mail reminders were sent to participants requesting they complete the follow-up survey at Timepoint 2.

### 2.5. Data Analysis

Data were exported into SPSS version 25.0 (SPSS, Chicago, IL, USA) which was used to provide a descriptive analysis of demographic details and to analyse the sexual attitudes and beliefs survey data. One-way repeat measures ANOVAs with within-subject effects were used to test for overall significant changes over time in total scores and scores for each statement. Pairwise comparisons were used to test for significant differences between each timepoint. A Bonferroni adjusted *p* value of 0.004 (0.05/12) was used to allow for multiple comparisons. Descriptive statistics were also used to summarise participant ratings of usefulness and utility.

## 3. Results

### 3.1. Participant Characteristics

Between January and June 2019, 44 participants completed the baseline survey (T0) and the follow-up questionnaires immediately after the e-learning (T1) and at 4 weeks (T2). The majority of participants were nurses working in urology and oncology settings with between five and twelve years of clinical experience. The most common reasons stated for undertaking the e-learning resource were to improve knowledge and communication skills when discussing sexual issues with patients. Participant demographics are summarised in Table 1.

### 3.2. Sexual Attitudes and Beliefs

The greatest potential barriers to sexual wellbeing discussions at baseline (T0) were with ‘I feel confident in my ability to address the sexual concerns of men living with prostate cancer’ (mean score/4 = 2.59: 95% CI = 2.40–2.73) and ‘I know the right language to use when discussing sexual concerns’ (mean score/4 = 2.52: 95% CI = 2.38–2.66) (See Table 2). The overall SABS scores on completion of the e-learning module did not change significantly from baseline. However, examination of individual statement scores revealed that there were significant within subject changes after completing the e-learning for key attitudes and beliefs. These included ‘awareness and understanding of how prostate cancer and its treatment might affect men’s sexual wellbeing’ (*F* = 23.657; *P* = 0.001) and ‘agreeing that sexual issues should be discussed with partners’ (*F* = 12.192; *P* = 0.001). In addition, participants were significantly more to agree that they had ‘I am confident in my ability to address sexual concerns’ (*F* = 27.351; *P* = 0.001) and ‘I know the right language to use when discussing sexual concerns’ (*F* = 27.013; *P* = 0.001) (See Table 2). 

Pairwise comparisons were used to identify where these significant differences were in terms of the different timepoints. This revealed significant differences in ‘understanding’ between T0 and T1 (mean score/4 = 3.18 vs. 3.79, respectively: *P* = 0.001) and between T0 and T2 (mean score/4 = 3.8 vs. 3.61, respectively: *P* = 0.001). For ‘confidence in ability to address sexual concerns’, significant differences were found between T0 and T1 (mean score/4 = 2.59 vs. 3.13, respectively: *P* = 0.001) but not between T0 and T2 (mean score/4 = 2.59 vs. 2.66, respectively: *P* = 0.063). Analysis of agreement with the statement that ‘sexual issues should be discussed with partners’ revealed significant differences between T0 and T1 (mean score/4 = 2.93 vs. 3.45, respectively: *P* = 0.001) and between T0 and T2 (mean score/4 = 2.93 vs. 3.39, respectively: *P* = 0.001). For the question on having the right language to use when discussing sexual concerns, significant differences were found between both T0 and T1 and T0 and T2 (mean score/4 = 2.52 vs. 2.98, respectively: *P* = 0.001 and mean score/4 = 2.52 vs. 2.85, respectively: *P* = 0.001).

### 3.3. Usefulness and Usability

Participant ratings on usefulness and usability immediately after completion revealed the e-learning resource was seen as containing relevant information (3.81/4; SD: 0.37) (see Table 3).

Free text comments on experience of using the resource were summarised into categories and identified that users found the interactive design to be the most frequent positive feature of the e-learning (*n* = 14/44: 31.8%). Other useful features identified were the addition of a practical framework to structure sexual wellbeing discussions (*n* = 13/44: 29.5%), the use of demonstration videos to help facilitate practice and rehearsal of conversations (*n* = 12/44: 27.3%) and inclusion of downloadable resources (*n* = 10/44: 22.7%). All participants (*n* = 44/44: 100%) accessed a link provided to an online version of the engagement framework included in the resource. The majority (*n* = 29/44: 65.9%) accessed at least one of the downloadable resources (for example, the patient handout). Suggested improvements to the resource included providing greater depth of information on sexual aids, stating recommended dosages where information on medications was included and developing the resource as a mobile app-based version. When asked as part of the survey completed at Timepoint 2 (4 weeks after completing the e-learning resource), over one-third (*n* = 14/44: 31.8%) provided examples where they had applied learning from the resource into their clinical practice. This included initiating discussions with patients they would not have previously spoken to about sexual wellbeing.

## 4. Discussion

This pilot study aimed to evaluate a brief e-learning resource that was designed to provide foundation level training to improve routine sexual wellbeing communication in prostate cancer care. Although overall SABS scores on completion of the e-learning did not change, significant improvements were observed for a number of individual survey item scores, indicating changes in important attitudes and beliefs. Baseline data indicated that participants regarded sexual wellbeing care and support as an important part of their professional role. However, confidence in their ability to address sexual concerns of men living with prostate cancer, and not knowing the right language to use when discussing sexual concerns, were both identified as potential barriers to discussing sexual wellbeing with patients (i.e., participants did not agree that they had confidence, or the right language to discuss sexual concerns), meaning that they had the greatest potential to improve following the e-learning resource. Completion of the e-learning resulted in significant improvements in both of these attitudinal factors, as well as in self-perceived knowledge and awareness of prostate cancers’ impact on sexual wellbeing. In addition, participants had a greater recognition that partners should be involved in sexual wellbeing discussions. An important finding was that improvement in perception of participants’ capacity to know the right language when discussing sexual wellbeing was maintained at least 1 month after completing the e-learning resource. These findings support the contention that brief e-learning can be an effective method of changing healthcare professionals’ attitudes and beliefs towards sexual wellbeing discussions in prostate cancer care. These effects were also similar to those observed in other studies examining more intensive, workshop-based interventions [21,22]. It is uncertain, however, whether changes in attitudes observed in this and in other studies would translate to longer-term changes in practice. For this to occur, such changes need to be maintained beyond the short-term period. Learning may need to be repeated and applied in everyday practice in order to facilitate measurable behavioural changes [23,24,25,26]. This contention is reinforced by findings from studies that suggest to address gaps between attitudes and actual clinical behaviour, additional training, including simulated practice and rehearsal, may be required [27,28]. Use of goal setting techniques included in this intervention were included to provide strategies to ensure discussions around sexual wellbeing become routine. However, goal setting techniques by themselves may not be sufficient to change practice [29,30]. One study, exploring the effects of a workshop-based intervention found participants infrequently discussed sexual wellbeing with patients as part of their practice after training, with most participants not reaching previously set targets [22].

There is evidence from existing studies to suggest that despite acknowledging that sexual care is part of their role there is often uncertainty over who is responsible for discussing sexual wellbeing issues, with healthcare professionals often being unaware of whether other colleagues discuss these issues with patients [31,32]. Healthcare professionals may also be uncertain as to when, and how to refer patients to other services, including sexual counselling. Level of discomfort has also been shown to be a significant predictor of addressing patients’ sexual concerns [33,34]. Such discomfort has been attributed to a fear of embarrassing patients or wanting to avoid “opening a can of worms” [35]. However, patients often expect healthcare professionals to raise the issue, and are comfortable with such conversations taking place [36,37]. To address these issues, interventions aimed at normalising sexual wellbeing conversations in clinical practice should be more widely implemented and should be a component of undergraduate healthcare professional training.

Previous sexual wellbeing communication training has frequently been based on existing sexual care models such as PLISSIT (Permission, Limited Information, Specific Suggestions, Intensive Therapy) [38,39,40,41,42]. This model places an emphasis on explicit ‘permission’ or approval to talk about sexual concerns at any point where it might be discussed and consider timing by ensuring patients are ‘ready’ to discuss sexual concerns. Furthermore, while educational interventions based on existing sexual care models such as PLISSIT and BETTER [43] provide well-developed and structured approaches, they often have limited active behaviour change components to support effective implementation into practice. These models may also include aspects that may present limitations to ensuring brief sexual care discussions are part of standard practice. These barriers can provide an ‘opt-out’ option leading to healthcare professionals not initiating discussions, potentially based on a perception that the patient is not ready or does not wish to discuss sexual issues [44]. While the EASSi engagement framework included in the e-learning draws on some aspects of these existing models, it attempts to build upon them by ensuring wider access to routine sexual care and support in prostate cancer care [45]. Its theoretical underpinning is more closely related to brief behaviour change models such as the 5 A’s model [46], which has been used as a framework to guide discussions in behavioural counselling interventions for smoking cessation and weight loss [47,48]. We postulate that addition of this simple, practical EASSi framework to guide and structure sexual wellbeing discussions may have resulted in the sustained improvement in participants’ perception that they had the right ‘language’ to use when discussing sexual concerns that was observed 1 month after completing the e-learning.

The intervention examined in this study can be used across settings and without specific training or expertise in sexual care counselling. In addition, it includes evidence-based behavioural change elements. The brevity of the framework and the combination of a routine assessment alongside provision of appropriate support also means it can be used at any stage in care, from pre-treatment to longer-term follow-up. Findings also indicate that the resource was seen as being easy to use and relevant to practice. The resource was used and viewed differently by various healthcare professionals in terms of its design, and function. This is in agreement with data from a recent systematic review exploring e-learning that concluded that effectiveness of training interventions can be influenced by learning style and mode of delivery [49]. e-learning is an increasingly substantial component of continuing professional development programmes for healthcare professionals. It offers numerous distinct advantages in comparison to face-to-face learning including wider reach, easier user access and improved usability [30,50]. e-learning resources have also been shown to have comparable efficacy and improved cost-effectiveness to traditional learning [51].

### Limitations

A number of limitations need to be taken into account when interpreting the findings of this study. The study did not include a control group or a randomised design. The sample size was also relatively small and comprised mainly of healthcare professionals from a nursing background. Furthermore, findings are based on a unidimensional outcome measure and analysis of individual survey items. This may reduce the ability to make strong conclusions on effectiveness of the e-learning resource and the generalisability of these findings. These factors should be considered in the design of further studies in this area.

## 5. Conclusions

Healthcare professionals should routinely engage with all patients to provide information and support to address and mange sexual wellbeing issues. However, existing communication and attitudinal barriers can limit this engagement and these barriers can be difficult to overcome. Providing healthcare professionals with only knowledge and awareness of the impact of prostate cancer and treatment on patient’s sexual wellbeing may not be sufficient to change practice. Including a practical framework to facilitate and structure sexual wellbeing conversations, and including behaviour change techniques such as reminders and prompts, may have contributed to the changes in important attitudes and beliefs around sexual wellbeing in this study that could support changes in clinical practice and behaviour. To support this, these tools should provide a practical resource to guide and support healthcare professionals to initiate sexual wellbeing discussions in routine clinical practice and should include tangible support in the form of downloadable materials to use in practice. Application of learning in practice may promote increased engagement around sexual wellbeing, ensuring fundamental but individualised support is provided for men and their partners. This has potential to address current gaps in care by addressing barriers to sexual wellbeing communication and providing a framework to promote routine delivery of essential sexual wellbeing support for men living with prostate cancer. Further studies are needed to explore the longer-term effectiveness of this e-learning resource and its impact on healthcare professional behaviour and patient important outcomes.

## Figures and Tables

**Table 1 ijerph-18-10045-t001:** Participant demographics (*n* = 44).

Participant Characteristic	Number (%)
Profession	Nurse: 31 (70.5%) Radiographer: 6 (13.6%) Doctor: 5 (11.4%) Cancer Support Worker: 2 (4.5%)
Years in practice	0–4 years: 7 (15.9%) 5–12 years: 32 (72.7%) More than 12 years: 5 (11.4%)
Days per week providing care for patients with prostate cancer	1–2 days per month: 8 (1.2%) 1–2 days per week: 24 (54.5%) 3–5 days per week: 12 (27.3%)
Previous training in sexual health or wellbeing communication	Yes: 8 (18.2%)
Primary reason for undertaking the e-learning	To improve communication skills: *n* = 13 (29.5%) To improve knowledge: *n* = 20 (45.5%) To support evidence-informed practice: *n* = 3 (6.8%) To increase confidence is discussion sexual wellbeing: *n* = 8 (18.2%)

**Table 2 ijerph-18-10045-t002:** Mean pre and post scores for sexual attitude and beliefs survey questions (*n* = 44).

				Within Subject Effects		Pairwise Comparisons	
Statement ***	T0 Mean (95% CI)	T1 Mean (95% CI)	T2 Mean (95% CI)	*F*	*P ^a^*	T0 vs. T1 *p ^a^*	T0 vs. T2 *p ^a^*
1. I understand how prostate cancer and its treatment might affect men’s sexual wellbeing	3.18 (2.98–3.41)	3.79 (3.55–3.82)	3.61 (3.31–3.62)	23.653	0.001 **	0.001 **	0.001 **
2. I am uncomfortable talking about sexual issues with men living with prostate cancer *	2.76 (2.69–3.14)	2.71 (2.54–2.83)	2.83 (2.73–3.06)	5.373	0.009	0.028	0.865
3. I feel confident in my ability to address the sexual concerns of men living with prostate cancer	2.59 (2.40–2.73)	3.13 (2.83–3.24)	2.66 (2.58–3.03)	27.355	0.001 **	0.001 **	0.063
4. Talking about sexual concerns with men living with prostate cancer can ‘open a can of worms’ *	2.82 (2.67–3.03)	2.69 (2.47–2.87)	2.78 (2.56–2.95)	2.670	0.082	0.175	0.564
5. Sexual concerns are an important topic to discuss with men living with prostate cancer	3.61 (3.47–3.73)	3.70 (3.57–3.83)	3.59 (3.45–3.74)	5.366	0.007	0.058	0.923
6. Discussing sexual concerns with men living with prostate cancer is part of my job	3.37 (3.14–3.44)	3.44 (3.33–3.64)	3.36 (3.21–3.52)	8.439	0.093	0.037	0.065
7. I make time to discuss sexual concerns with men living with prostate cancer	2.98 (2.64–3.21)	2.86 (2.78–3.13)	3.85 (2.81–3.16)	12.292	0.001 **	0.034	0.0013 **
8. Sexual issues should be discussed only if initiated by men living with prostate cancer *	3.26 (3.08–3.37)	3.31 (3.22–3.47)	3.19 (3.04–3.49)	8.261	0.034	0.878	0.065
9. I find it difficult to talk to older men living with prostate cancer about sexual concerns *	2.88 (2.76–3.15)	2.81 (2.65–2.97)	2.89 (2.72–3.07)	4.594	0.017	0.048	0.429
10. I know the right language to use when discussing sexual concerns with men living with prostate cancer	2.52 (2.38–2.66)	2.98 (2.78–3.13)	2.85 (2.64–2.93)	27.013	0.001 **	0.001 **	0.001 **
11. Sexual issues should be discussed with partners of men living with prostate cancer	2.93 (2.91–3.22)	3.45 (3.22–3.54)	3.39 (3.14–3.39)	12.192	0.001 **	0.001 **	0.001 **
12. Men living with prostate cancer do not expect healthcare professionals to ask about sexual concerns *	2.24 (2.07–3.15)	2.21 (2.11–3.05)	2.64 (2.46–3.11)	1.395	0.253	0.695	0.137
Total score/48(95% CI)	35.14 (34.82–36.87)	37.08 (36.12–38.41)	37.64 (35.72–37.75)	3.145	0.060	0.134	0.587

T0 = Timepoint 0 (baseline). T1 = Timepoint 1 (immediately post completion). T2 = Timepoint 2 (1-month post completion). *** Modified sexual attitudes and beliefs questionnaire consists of 12 statements scored/4 based on the following criteria: 1 = Strongly Disagree; 2 = Disagree; 3 = Agree; 4 = Strongly Agree. A total score is assigned/48. ** Indicates a significant effect. * indicates a score which is reversed for data analysis. *^a^* Bonferroni-adjusted *P* value for multiple comparisons: *P* < 0.004. CI = confidence interval. Boxplots showing the distribution of scores for each statement are shown in Appendix A.

**Table 3 ijerph-18-10045-t003:** Participant views on usefulness and usability recorded following completion of the e-learning resource (*n* = 44).

Statement	Mean/4 (SD)
1. The e-learning resource included information that will be useful for my practice	3.81 (0.37)
2. I would recommend others use the e-learning module	3.01 (0.40)
3. I will use the e-learning resource as a resource	3.53 (0.42)
4. I thought the e-learning resource was easy to use	3.42 (0.54)

Each statement is scored/4 based on the following criteria: 1 = Strongly Disagree; 2 = Disagree; 3 = Agree; 4 = Strongly Agree. SD = Standard Deviation.

## Data Availability

The data presented in this study are available on request from the corresponding author. The data are not publically available due to their containing information that could compromise the privacy of research participants.

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
