# Peer review of "Effects of a Brief E-Learning Resource on Sexual Attitudes and Beliefs of Healthcare Professionals Working in Prostate Cancer Care: A Pilot Study"

_ijerph, 2021, doi:10.3390/ijerph181910045_

Round 1
Reviewer 1 Report
Report on “Effects of a brief e-learning resource on sexual attitudes and beliefs of healthcare professionals working in prostate cancer care: A single arm pre and post‐test study” by Eilis M McCaughan et al.
The manuscript studies the effect on healthcare professionals of a brief e-learning resource designed to improve sexual wellbeing support. I think that the methodology is quite clear and the manuscript it is well written, but I have serious concerns about the lack of information provided by authors. Indeed, the topic is important in urology as follow-up of patients is a crucial issue in prostate cancer.
My concerns lie in the lack of information about the e-learning platform, the analysis of the reliability of the questionnaire and some minor point of the methodology:
Concerns:
- The authors provide the general link of moodle software, instead of providing a link to the platform or package that they have developed for the e-learning resource. It is difficult to evaluate a tool that it is not accessible to be reviewed. I recommend authors to provide a temporal access with an invitation role to review the e-learning resource. In addition, no information appears to know if this resource is going to be freely available or not (A link to the e-learning resource was also included on the healthcare professional online learning platform of Prostate Cancer UK). This is an important point to judge the interest of the study.
- As the authors declare in the limitations section, there was no randomized design in the study. In my opinion, this is very common in the first analysis of a questionnaire, but as a consequence, this work has the structure of a pilot study rather than a generalizable study. I recommend authors to include the character of a “pilot study” in the title and material and method section.
- The reliability of the questionnaire should be analyzed. The alpha Cronbach value for the questionnaire must be provided. Additionally, the correlation of each item with the total can be reported.
- The results are reported by mean and standard deviation, although the normal hypothesis of the results was not analyzed. In my opinion, results can be complemented by a panel of boxplots that compare the results of the pre and post tests for the 12 items analyzed giving a non-parametric comparison of the results. Probably 36 boxplots could be too information for a unique panel, therefore you can distribute boxplots in several panels.
Author Response
Thank you for the your comments which were very helpful

Reviewer 2 Report
The authors of this article address an interesting issue that is valuable to health care practice. The issue is important both from the perspective of professional activity of health professionals and students of medical sciences. The article meets all the requirements of a scientific paper. The research is embedded in the quantitative model and meets the standards of the quantitative process. The following remarks may contribute to improving the readability of the article for potential recipients.
Keywords: I propose to add the word "healthcare professionals".
Introduction: it would be helpful to specify which healthcare professionals the authors have in mind, for example by describing their role in supporting patients in the area of their sexual life.
Materials and methods: please specify the details of the SABS scale modification and the psychometric properties of the tool used in the presented research.
It would also be useful to provide indicators of the analyzed attitudes and beliefs. I notice some ambiguity in the description of the variables: the aim of the research is to find out professionals’ attitudes and beliefs, and in the description of the tool we read: perceived/felt (?) barriers to sexual wellbeing communications.
Results: Table 1 needs to be improved so that information in both columns aligns
Discussion:
(1) Please make a clear note that the changes were not observed in the total score but only in a few items. Perhaps the scale used by the authors is not appropriate for the job responsibilities and capabilities of such a diverse group of professionals.
(2) It is worth adding that Healthcare Professionals operate in a specific institutional context that provides guidelines for carrying out professional tasks. Moreover, they collaborate/should collaborate with professionals from other disciplines who may be more competent to support patients with sexual life problems (psychologists, sexologists). The authors allude in part to this issue when they write about the dilemmas faced by professionals (p. 7) who are unsure where the responsibility lies.
(3) It is worth adding that training in the area of sexuality counseling should be an integral part of medical student training.
Limitations:
(1) A relatively small group (even though the authors calculated its minimum size).
(2) The scale, which has only one dimension, and the analysis of results based on items.
(3) Diversity of the group of professionals who perform different professional tasks.
Author Response

(The authors gave the same response as above.)

Round 2
Reviewer 1 Report
I appreciate the effort of authors to improve the manuscript, all my concerns have been amended.